# Identification of TAPBPL as a novel negative regulator of T-cell function

Yujun Lin[1,2] (iD), Cheng Cui[1], Min Su[1], Lawrence K Silbart[1], Haiyan Liu[1,3], Jin Zhao[1], Lang He[1,4], Yuanmao Huang[1,5], Dexin Xu[1,6], Xiaodan Wei[1,7], Qian Du[8] & Laijun Lai[1,9,*]

## Abstract

T cell stimulatory and inhibitory molecules are critical for the regulation of immune responses. In this study, we identify a novel T cell co-inhibitory molecule TAPBPL, whose amino acid sequence shares homology with known B7 family members. TAPBPL protein is expressed on resting and activated T cells, B cells, monocytes, and dendritic cells (DCs), as well as on some tumor tissues. The putative TAPBPL receptor is expressed on activated CD4 and CD8 T cells. A soluble recombinant human TAPBPL-IgG Fc (hTAPBPL-Ig) fusion protein inhibits the proliferation, activation, and cytokine production of both mouse and human T cells *in vitro*. *In vivo* administration of hTAPBPL-Ig protein attenuates experimental autoimmune encephalomyelitis (EAE) in mice. Furthermore, an anti-TAPBPL monoclonal antibody neutralizes the inhibitory activity of hTAPBPL-Ig on T cells, enhances antitumor immunity, and inhibits tumor growth in animal models. Our results suggest that therapeutic intervention of the TAPBPL inhibitory pathway may represent a new strategy to modulate T cell-mediated immunity for the treatment of cancer, infections, autoimmune diseases, and transplant rejection.

**Keywords** autoimmune disease; B7 family; T cells; TAPBPL; tumor immunity
**Subject Category** Immunology

## Introduction

T cells play critical roles in the adaptive immune system, protecting the human body against cancer, bacterial, viral, fungal, and parasitic infections. In order to elicit protective immunity to cancer and infection, and to prevent an overactive immune response that can lead to autoimmune disease, T cell immune responses have to be tightly controlled.

The B7 family plays a key role in controlling immune responses and belongs to the immunoglobulin (Ig) superfamily. A number of B7 family ligands have been identified, such as B7-1 (CD80), B7-2 (CD86) (Freeman *et al*, 1991; Freeman *et al*, 1993), PD-L1 (B7-H1) (Dong *et al*, 1999; Freeman *et al*, 2000), PD-L2 (B7-DC) (Latchman *et al*, 2001; Tseng *et al*, 2001), B7-H2 [inducible T cell co-stimulator ligand (ICOS)] (Swallow *et al*, 1999; Yoshinaga *et al*, 1999; Ling *et al*, 2000; Wang *et al*, 2000), B7-H3 (Chapoval *et al*, 2001), B7-H4 (B7x, B7S1) (Prasad *et al*, 2003; Sica *et al*, 2003; Zang *et al*, 2003), B7-H5 (HHLA2) (Zhao *et al*, 2013; Zhu *et al*, 2013), and B7-H6 (Brandt *et al*, 2009). Several new drugs targeting the B7 family ligands or their counterreceptors have been approved by the FDA for the treatment of cancer, autoimmune disease, and transplantation rejection. For example, blocking antibodies against PD-L1 or its receptor PD-1, or the B7-1 and B7-2 inhibitory receptor CTLA-4 have been used in the treatment of cancer patients. Conversely, recombinant CTLA-4-Fc fusion protein has been approved by the FDA to treat rheumatoid arthritis and to prevent kidney transplantation rejection.

Because of the potential clinical applications, there has been intense interest in identifying additional T-cell regulators. In this study, we identify an antigen processing (TAP) binding protein like (TAPBPL)/TAP binding protein-related (TAPBPR) molecule as a novel B7 family-related molecule. TAPBPL/TAPBPR was originally identified on chromosome position 12p13.3 near an MHC paralogous locus and was known for involvement in peptide selection (Du Pasquier, 2000; Teng *et al*, 2002; Hermann *et al*, 2015; Morozov *et al*, 2016; Neerincx & Boyle, 2017). We found that TAPBPL shares a significant sequence similarity with some known B7 family members. TAPBPL protein is also expressed on the surface of T cells, and antigen-presenting cells (APCs) including resting B cells, monocytes, macrophages, and DCs, as well as on some cancer cells including leukemia cells. A

---

1   Department of Allied Health Sciences, University of Connecticut, Storrs, CT, USA
2   The Second Affiliated Hospital of Fujian Medical University, Quanzhou, China
3   Shandong Provincial Hospital Affiliated to Shandong First Medical University, Shandong, China
4   School of Biological Science and Technology, Chengdu Medical College, Chengdu, China
5   Zhangzhou Affiliated Hospital of Fujian Medical University, Zhangzhou, China
6   Fuzhou Pulmonary Hospital, Fuzhou, China
7   College of Basic Medicine, Binzhou Medical University, Yantai, China
8   Plant Biology Section, Cornell University, Ithaca, NY, USA
9   University of Connecticut Stem Cell Institute, University of Connecticut, Storrs, CT, USA
    *Corresponding author. Tel: +1 860 486 6073; Fax: +1 860 486 0534; E-mail: laijun.lai@uconn.edu

soluble TAPBPL-Ig fusion protein inhibits the proliferation and activation of CD4 and CD8 T cells *in vitro* and ameliorates autoimmune disease EAE *in vivo*. In contrast, anti-TAPBPL antibody enhances antitumor immunity and inhibits tumor growth *in vivo*. Therefore, TAPBPL contains typical features of B7 family members, suggesting that it is a B7 family member or a B7 family-related molecule.

# Results

## TAPBPL shares sequence and structural similarities with existing B7 family members

B7-H5/HHLA2 is a member of the B7 family and shares 10–18% amino acid identity to other B7 molecules (Zhao *et al*, 2013; Zhu *et al*, 2013). Functionally, B7-H5/HHLA2 inhibits the proliferation and cytokine production of T cells (Zhao *et al*, 2013; Zhu *et al*, 2013). Through a series of genome-wide database searches, we found that human TAPBPL has 15% identity and 16% similarity in amino acid with B7-H5/HHLA2 (Fig 1A). TAPBPL also shares 10–14% identity with some members of the B7 family (Fig 1A).

Like the B7 family molecules, TAPBPL is a member of the Ig superfamily (Du Pasquier, 2000; Teng *et al*, 2002; Hermann *et al*, 2015; Morozov *et al*, 2016). The TAPBPL gene encodes a signal peptide region in the N terminus, an extracellular region, a transmembrane domain, and an intracellular region (Fig 1B). The B7 family members typically contain IgV and IgC domains in the extracellular portion. The extracellular region of TAPBPL also contains an lgV domain and an IgC domain (Fig 1B). TAPBPL is conserved among vertebrates, and human TAPBPL (hTAPBPL) and mouse TAPBPL (mTAPBPL) proteins have 69% homology in amino acid sequences (Teng *et al*, 2002; Hermann *et al*, 2015).

## TAPBPL protein is expressed on the cell surface of APCs and T cells, and on some tumor tissues

We first generated hTAPBPL-Ig and mouse TAPBPL (mTAPBPL)-Ig fusion proteins by cloning the extracellular domain of the hTAPBPL gene into an expression vector containing signal sequences and the constant region of mouse IgG2a. The vector was transfected into HEK-293 cells to produce a recombinant hTAPBPL-Ig or mTAPBPL-Ig fusion protein. We then purified the fusion proteins from the supernatant of HEK-293 cells. A relative high purity of TAPBPL-Ig fusion protein was obtained as shown by SDS–PAGE and confirmed by Western blot using anti-mouse IgG2a and TAPBPL antibodies (Appendix Fig S1A).

We then produced anti-hTAPBPL monoclonal antibodies (mAbs) by immunizing BALB/c mice with hTAPBPL-Ig protein. The splenocytes were fused to X63-Ag8.653 myeloma cells to produce hybridomas. ELISA was performed to screen hybridomas secreting mAbs which bound to the hTAPBPL-Ig fusion protein but not with control Ig. We obtained an anti-hTAPBPL mAb (clone 54) that reacted with hTAPBPL-Ig and mTAPBPL-Ig, but not with control Ig protein (Appendix Fig S1B). Furthermore, the anti-hTAPBPL mAb stained P388 leukemia cells, but not mTAPBPL siRNA-treated P388 cells (Appendix Fig S1C), confirming the specificity of the mAb.

Since most of the B7 family molecules are expressed on APCs and/or T cells, we assessed TAPBPL protein expressed on these cells. As shown in Fig 2A and B, TAPBPL was expressed on the cell surface of mouse CD4$^+$ and CD8$^+$ T cells at low levels. The expression level of TAPBPL on CD8$^+$ T cells was slightly increased upon activation by anti-CD3 and anti-CD28 antibodies. TAPBPL protein was also detected on APCs including CD11b$^+$ monocytes, F4/80$^+$ macrophages, CD11c$^+$ dendritic cells (DCs), and B220$^+$ and/or CD19$^+$ B cells. Furthermore, the expression levels of TAPBPL on monocytes and DCs were increased upon activation by LPS or IFN-γ (Fig 2A and B and Fig EV1A), whereas those on macrophages and B cells were not changed upon activation (Fig 2A and B). We also used qRT–PCR to detect the TAPBPL mRNA expression levels in the immune cells. The results were consistent with the TAPBPL protein expression levels on these cells (Fig EV1B). Taken together, our results suggest that TAPBPL is constitutively expressed on the surface of APCs and T cells and that expression is upregulated on monocytes, DCs cells, and CD8 T cells after activation.

Since PD-L1 is a critical B7 family T cell inhibitory molecule, we determined the co-expression of TAPBPL and PD-L1 on mouse B cells, DCs, and macrophages. Some of these cells co-expressed PD-L1 and TAPBPL, while a proportion of these cells expressed TAPBPL alone, suggesting that targeting TAPBPL and PD-L1 in patients may have a synergetic effect (Fig EV1C).

We also analyzed the expression of TAPBPL on human peripheral blood mononuclear cells (PBMCs). Human monocytes and B cells were activated with LPS/IFN-γ for 3 days as described (Zhao *et al*, 2013). Immature DCs were generated from blood monocytes by incubation with GM-CSF/IL-4 and were induced to become mature DCs with LPS/IFN-γ (Zhao *et al*, 2013). TAPBPL was expressed on resting and activated CD19$^+$ B cells, CD14$^+$ monocytes, and mature DCs but not on immature DCs (Fig EV1D and E).

We then assessed the expression of TAPBPL protein in human normal and tumor tissues by immunohistochemistry. As shown in Fig 2C, TAPBPL protein was detected in normal breast, colon, liver, lung, and prostate tissues at low levels, as compared to isotype antibody staining. TAPBPL protein was expressed in liver, lung, and prostate cancer tissues at medium to high levels (Fig 2D). Therefore, the expression of TAPBPL in liver, lung, and prostate cancer tissues was higher than that observed on matching normal tissues (Fig 2C and D). TAPBPL protein was largely located on the plasma membrane and cytoplasm of the cancer cells.

We also examined TAPBPL protein expression on cancer cell lines by flow cytometry. As shown in Fig 2E and F, TAPBPL was expressed highly on the cell surface of murine neuro-2a neuroblastoma and P388 leukemia cells, and weakly on murine Lewis lung carcinoma, CT-26 colon cancer, and B16F10 melanoma cells. The TAPBPL protein expression levels on the cancer cells were also consistent with the TAPBPL mRNA expression levels in these cells (Fig EV1F). Furthermore, we analyzed the expression of TAPBPL on cancer cells following IFNγ stimulation *in vitro* and found that the expression levels of TAPBPL on neuro-2a neuroblastoma and B16F10 melanoma were upregulated upon stimulation (Fig EV1G and H).

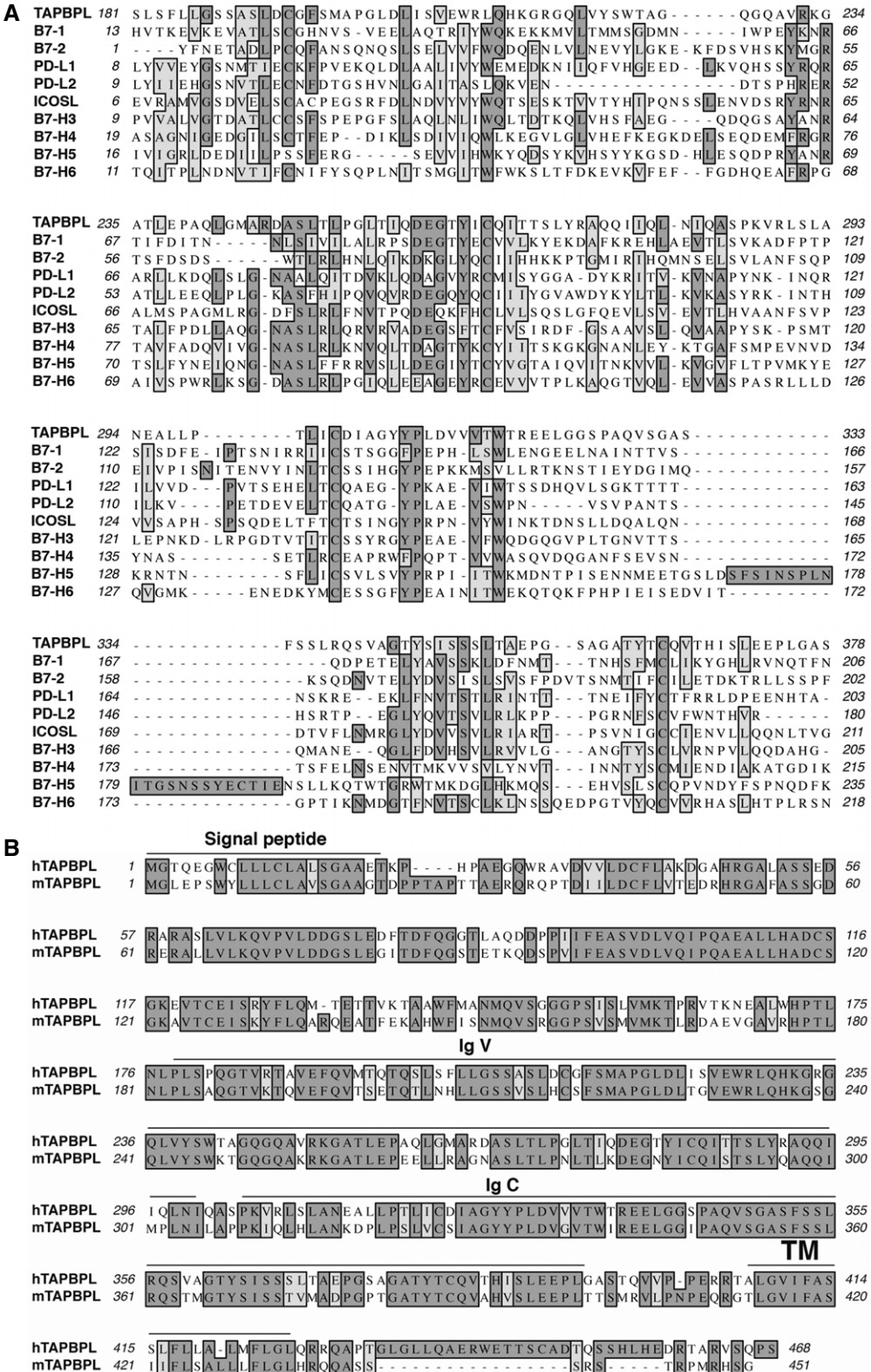

**Figure 1. Characterization of TAPBPL amino acids.**

A  Alignment of the extracellular region of human TAPBPL (hTAPBPL) with that of some known human B7 family members. Identical amino acids are shaded black. Amino acids with strong homologies are shaded in gray.

B  Alignment of hTAPBPL with mouse TAPBPL (mTAPBPL). Predicted signal peptide, IgV- and IgC-like, and transmembrane (TM) regions for hTAPBPL are marked.

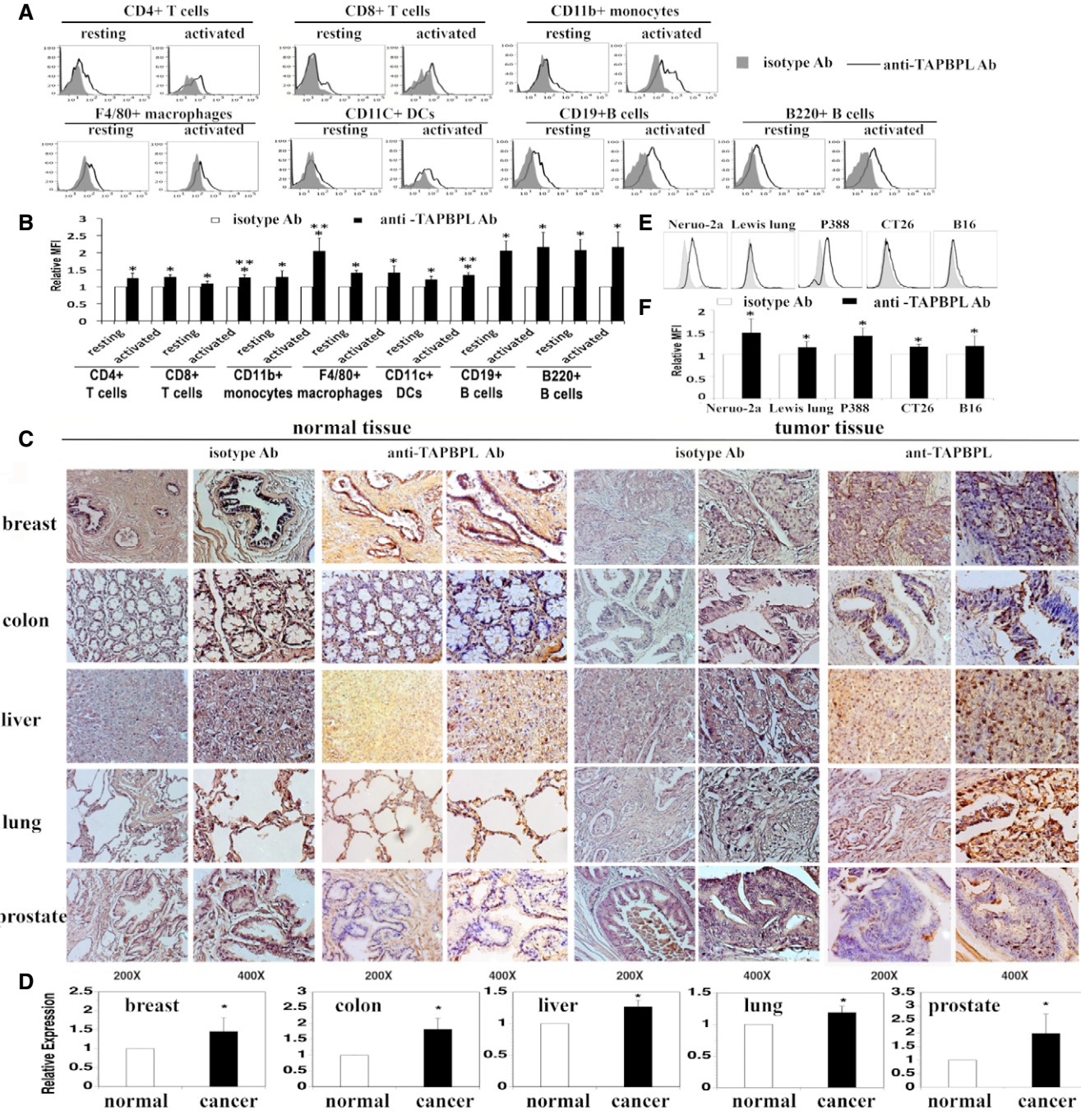

**Figure 2. The expression patterns of TAPBPL protein.**

A, B  Analysis of TAPBPL protein expression on immune cells. Splenocytes from C57BL/6 mice were freshly harvested and analyzed for TAPBPL protein expression on resting immune cells. To obtain activated T cells, the splenocytes were incubated with anti-CD3 (1 μg/ml) and anti-CD28 (0.5 μg/ml) antibodies for 3 days. To obtain activated B cells, DCs, monocytes, and macrophages, the splenocytes were incubated with LPS (10 μg/ml) for 3 days. The resting and activated immune cells were stained with anti-TAPBPL or isotype antibody (Ab), as well as anti-CD4, CD8, CD11b, F4/80, CD11c, CD19, or B220 antibody to identify immune cells. (A) Representative flow cytometric profiles and (B) statistical analysis showing the expression levels of TAPBPL protein on resting and activated immune cells (n = 3). Significance was calculated by two-way ANOVA with Tukey test. *P < 0.05 compared with isotype Ab; **P < 0.05 compared with resting cells.

C  Determination of TAPBPL protein expression in normal and tumor human tissues by immunohistochemistry using anti-TAPBPL or isotype Ab.

D  Statistical analysis of the expression levels of TAPBPL on tumor tissues. The data were presented as relative expression levels of TAPBPL on tumor tissues versus relative normal tissues (each group was normalized with isotype Ab first) (n = 5). Significance was calculated by two-tailed Student's t-test. *P < 0.05 compared with normal tissue.

E, F  Analysis of TAPBPL protein expression on the indicated cancer cell lines. (E) Representative flow cytometric profiles and (F) statistical analysis showing the expression levels of TAPBPL protein on the cancer cells (n = 3). Significance was calculated by two-tailed Student's t-test. *P < 0.05 compared with isotype Ab.

### The expression of the putative TAPBPL receptor

To determine the expression pattern of the putative TAPBPL receptor, TAPBPL-Ig and control Ig proteins were biotinylated.

Splenocytes from C57BL/c mice were stained with the biotinylated proteins, followed by streptavidin-PE. Flow cytometric analysis showed that TAPBPL-Ig scarcely bound to resting CD4$^+$ and CD8$^+$ T cells; however, the binding increased significantly when CD4$^+$ and

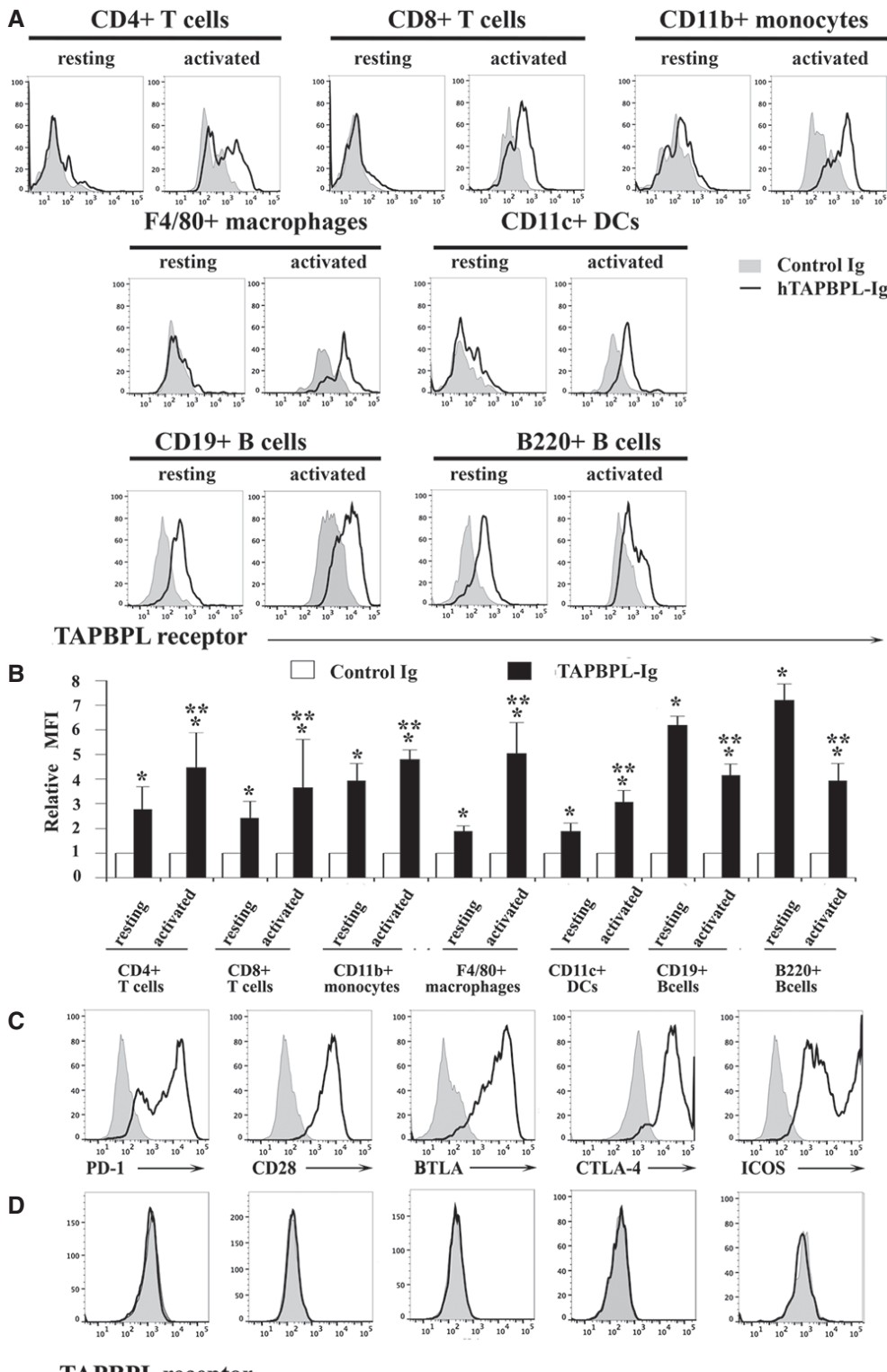

**Figure 3.**

◀

**Figure 3. The expression pattern of the putative TAPBPL receptor.**

A, B Splenocytes from C57BL/6 mice were freshly harvested. Resting and activated T cells, monocytes, macrophages, DCs, and B cells were obtained as in Fig 2. The resting and activated immune cells were stained with biotinylated TAPBPL-Ig or control Ig, followed by streptavidin-PE, as well as anti-CD4, CD8, CD11b, F4/80, CD11c, B220, or CD19 antibody to identify immune cells. (A) Representative flow cytometric profiles and (B) statistical analysis showing the binding TAPBPL-Ig or control Ig to freshly harvested and activated immune cells ($n = 3$). Significance was calculated by two-way ANOVA with Tukey test. *$P < 0.05$ compared with control Ig. **$P < 0.05$ compared with resting cells.

C HEK-293 cells were transfected with an expression vector containing the murine PD-1, CD28, BTLA, CTLA-4, or ICOS gene and screened for cells stably expressing each gene. The transfected cells were stained with antibodies against the respective PD-1, CD28, BTLA, CTLA-4, or ICOS protein (open histograms) or isotype Ab (shaded histograms). Representative flow cytometric profiles showing the expression of each receptor.

D PD-1, CD28, BTLA, CTLA-4, or ICOS gene-transfected HEK-293 cells were stained with biotinylated TAPBPL-Ig and control Ig, followed by streptavidin-PE. Representative flow cytometric showing the binding of TAPBPL-Ig (open histograms) or control Ig protein (shaded histogram) to the transfected cells. The data are representative of 3 independent experiments.

CD8$^+$ T cells were activated by anti-CD3 and anti-CD28 antibodies (Fig 3A and B and Fig EV2).

We also analyzed the expression of the putative TAPBPL receptor on other immune cells. Although TAPBPL-Ig bound weekly to resting CD11b$^+$ monocytes, F4/80$^+$ macrophages, and CD11c$^+$ DCs, binding was increased upon activation (Fig 3A and B and Fig EV2). TAPBPL bound to resting B220$^+$ B cells, but the binding decreased when the B cells were activated by LPS (Fig 3A and B).

To determine whether TAPBPL binds to molecules previously identified as receptors of the known B7 family members, HEK-293 cells were transfected with an expression vector containing the PD-1, CD28, BTLA, CTLA-4, or ICOS gene. The expression of these receptors on the transfected 293 cells was confirmed by flow cytometric analysis with the antibodies against the respective receptors (Fig 3C). The binding of TAPBPL to the transfected HEK-293 cells was then analyzed. As shown in Fig 3D, TAPBPL did not bind to the PD-1, CD28, BTLA, CTLA-4, or ICOS gene-transfected cells.

Taken together, our results suggest that the putative TAPBPL receptor is expressed on activated CD4 and CD8 T cells, monocytes, macrophages, and DCs. The receptor is also expressed on resting B cells, and the expression level is downregulated when B cells are activated. The putative TAPBPL receptor seems to be distinct from PD-1, CD28, BTLA, CTLA-4, and ICOS.

**TAPBPL inhibits T cell activation *in vitro***

Since TAPBPL shares sequence homology with the B7 family members, TAPBPL is expressed on APCs, and the TAPBPL putative receptor is expressed on T cells, we hypothesized that TAPBPL has a regulatory role on T cells. We first determined whether hTAPBPL-Ig affected T cell activation *in vitro*. After splenocytes were stimulated with anti-CD3 antibody in the presence of hTAPBPL-Ig or control Ig, T cells were analyzed for the expression of CD69, an early activation marker, 24 h later. Since the molecular weight of hTAPBPL-Ig is ~2.7-fold higher than that of control Ig, we used equimolar amounts of recombinant mouse IgG2a (control Ig) protein as a control. As shown in Fig 4A and B, hTAPBPL-Ig reduced anti-CD3-activated CD69 expression on both CD4 and CD8 T cells. Similarly, the expression levels of CD69 on CD4 and CD8 T cells activated by both anti-CD3 and anti-CD28 antibodies were also significantly decreased by hTAPBPL-Ig (Fig 4C and D). We also determined the ability of mTAPBPL-Ig to affect mouse T cell activation *in vitro*. mTAPBPL-Ig, when used at lower concentrations than hTAPBPL-Ig (0.8–1.6 vs. 5–15 μg/ml), could inhibit anti-CD3-activated CD69 expression on both CD4 and CD8 T cells (Appendix Fig S2A and B).

T cells can be divided into naïve (CD44$^{lo}$CD62L$^{hi}$) and effector memory (CD44$^{hi}$CD62L$^{lo}$) T cells based on the expression levels of CD44 and CD62L. We next analyzed the effect of hTAPBPL-Ig on these T-cell subsets. We found that the percentages and numbers of CD44$^{hi}$CD62L$^{lo}$ CD4 and CD8 effector memory T cells were significantly lower in the presence of hTAPBPL-Ig than those in the control group when T cells were stimulated by either anti-CD3 antibody or anti-CD3 plus anti-CD28 antibodies (Fig 4E–I and K). In contrast, the percentages and numbers of CD44$^{lo}$CD62L$^{hi}$ CD4 and CD8 T naïve cells were significantly higher in the presence of hTAPBPL-Ig (Fig 4E–G and I–L). The results further suggest that hTAPBPL-Ig inhibits the activation of CD4 and CD8 T cells. Taken together, our results suggest that TAPBPL-Ig inhibits TCR-mediated activation of both CD4 and CD8 T cells *in vitro*.

**TAPBPL inhibits T-cell proliferation and cytokine production *in vitro***

We then determined whether TAPBPL-Ig protein affected T cell proliferation. CD3$^+$ T cells were purified from splenocytes of C57BL/c mice and cultured on plates pre-coated with anti-CD3 antibody in the presence of hTAPBPL-Ig or control Ig. Three days later, T cell proliferation was measured by [$^3$H] thymidine incorporation. As shown in Fig 5A, TAPBPL-Ig inhibited anti-CD3-induced T cell proliferation in a dose-dependent manner, with ~70% and 81% inhibition in the presence of 10 and 15 μg/ml hTAPBPL-Ig, respectively, as compared to equimolar amount of control Ig. We also determined whether hTAPBPL-Ig could inhibit ani-CD3 and anti-CD28 antibody-induced T cell proliferation. Similarly, hTAPBPL-Ig inhibited the T cell proliferation in a dose-dependent manner, with ~66% and 70% inhibition in the presence of 10 and 15 μg/ml hTAPBPL-Ig, respectively (Fig 5B).

To confirm the inhibitory effect on T cell proliferation and to determine whether hTAPBPL-Ig inhibits CD4 and/or CD8 T cells, splenocytes were labeled with carboxyfluorescein diacetate succinimidyl ester (CFSE) and cultured with anti-CD3 antibody and graded doses of hTAPBPL-Ig or equimolar amounts of control Ig. T cell proliferation was analyzed by determining CFSE fluorescent intensity in CD4$^+$ or CD8$^+$ T cells by flow cytometry. Consistent with results from the [$^3$H] thymidine incorporation assay, hTAPBPL-Ig inhibited anti-CD3-induced proliferation of both CD4$^+$ and CD8$^+$ T cells (Fig 5C–E). We then determined the ability of mTAPBPL-Ig to affect mouse T cell proliferation. Similarly, mTAPBPL-Ig inhibited anti-CD3-induced CD4 or CD8 T cell proliferation at lower concentrations than hTAPBPL-Ig (Appendix Fig S2C and D).

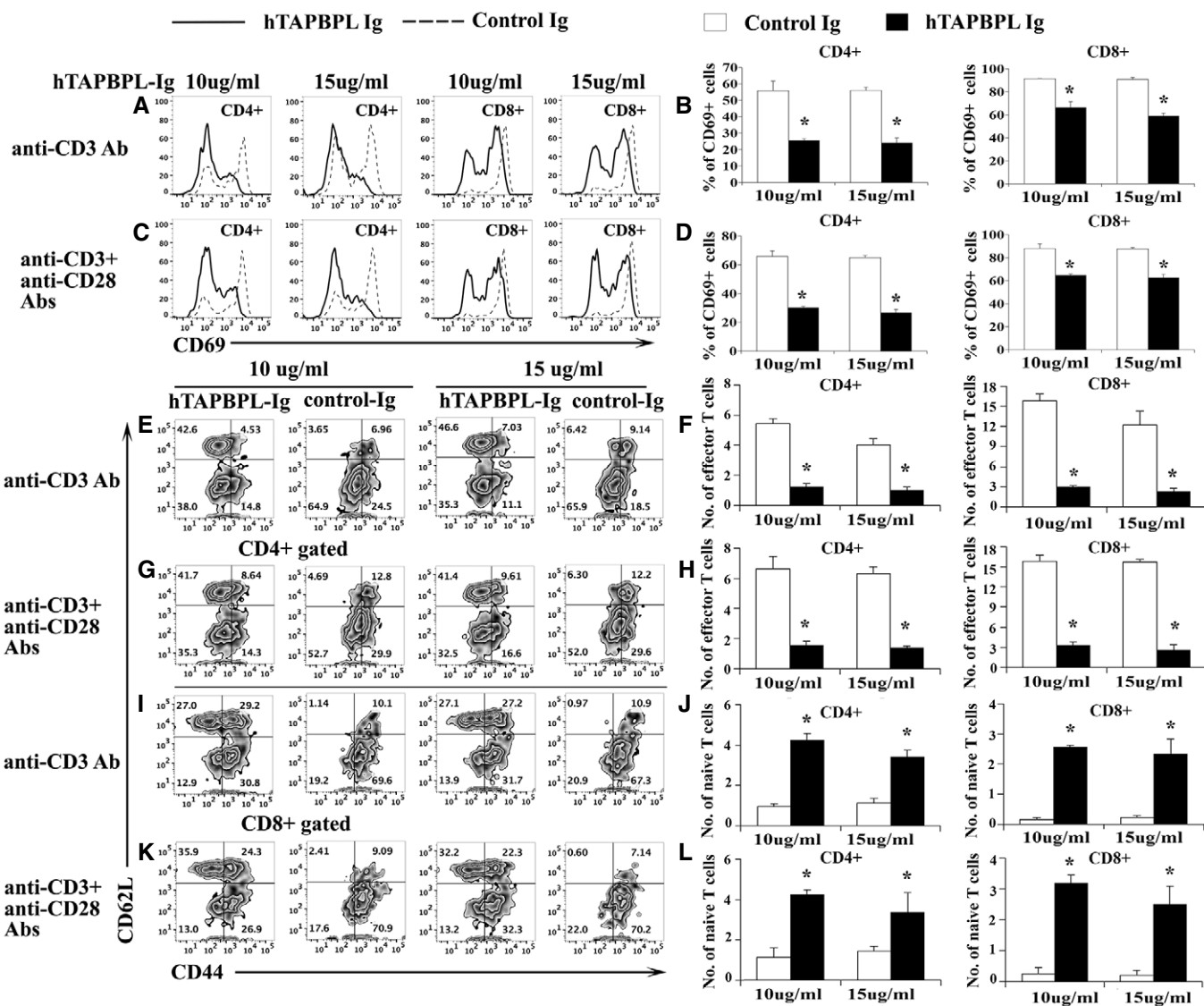

**Figure 4. The effect of hTAPBPL-Ig protein on T cell activation *in vitro*.**

A–L Splenic cells from C57BL/6 mice were cultured with (A, B, E, F, I, J) anti-CD3 antibody (1 μg/ml) or (C, D, G, H, K, L) anti-CD3 (1 μg/ml) and anti-CD28 (0.5 μg/ml) antibodies in the presence of graded doses of hTAPBPL-Ig (10 and 15 μg/ml) or equimolar amounts of control Ig (3.75 and 5.63 μg/ml) for (A-D) 1 day or (E-L) 3 days. The cells were analyzed for (A–D) CD69⁺, (E-I, K) CD44^hiCD62L^lo cells, and (E, G, I–L) CD44^loCD62L^hi cells in CD4 and CD8 T cells. (A, C, E, G, I, K) Representative flow cytometric and (B, D, F, H, J, L) statistical analyses of the (B, D) percentages of CD69⁺, and the numbers (X10⁵) of (F, H, J, L) CD44^hiCD62L^lo effector memory T cells and CD44^loCD62L^hi naïve T cells in CD4 and CD8 T cells. The data are expressed as mean + SD (*n* = 3). Significance was calculated by two-tailed Student's *t*-test. *$P < 0.05$ compared with control Ig.

Having shown that hTAPBPL-Ig inhibits murine T cell proliferation *in vitro*, we next determined whether hTAPBPL-Ig also inhibits the proliferation of human T cells. Purified human T cells were cultured with anti-human CD3 antibody in the presence of graded doses of hTAPBPL-Ig or control Ig for 3 days. T cell proliferation was measured by [³H] thymidine incorporation. As shown in Fig 5F, hTAPBPL-Ig significantly inhibited the proliferation of human T cells. When compared to the doses of hTAPBPL-Ig that influences murine T cells, the doses for human T cells were lower (Fig 5F vs. A).

We also examined whether hTAPBPL-Ig affects cytokine production from T cells *in vitro*. Murine purified T cells were stimulated with anti-CD3 antibody in the presence of hTAPBPL-Ig or control Ig

protein for 3 days. The cytokine-producing CD4 T cells were analyzed by flow cytometry. hTAPBPL-Ig significantly reduced the percentages of IFNγ- or IL-17A-producing (but not GM-CSF- or TNFα-producing) CD4⁺ T cells (Appendix Fig S3A and B). Collectively, our results suggest that TAPBPL-Ig inhibits the activation, proliferation, and certain cytokine production of T cells *in vitro*.

To investigate the mechanisms by which TAPBPL affects T cells, we analyzed and compared TCR-induced signaling molecules in TAPBPL- and PD-L1-treated T cells. Purified mouse CD3 T cells were stimulated with anti-CD3 and anti-CD28 antibodies in the presence of hTAPBPL-Ig, PD-L1-Ig, or control Ig. Since AKT, p38, and JNK are involved in multiple cellular processes (Su & Karin, 1996; Wang

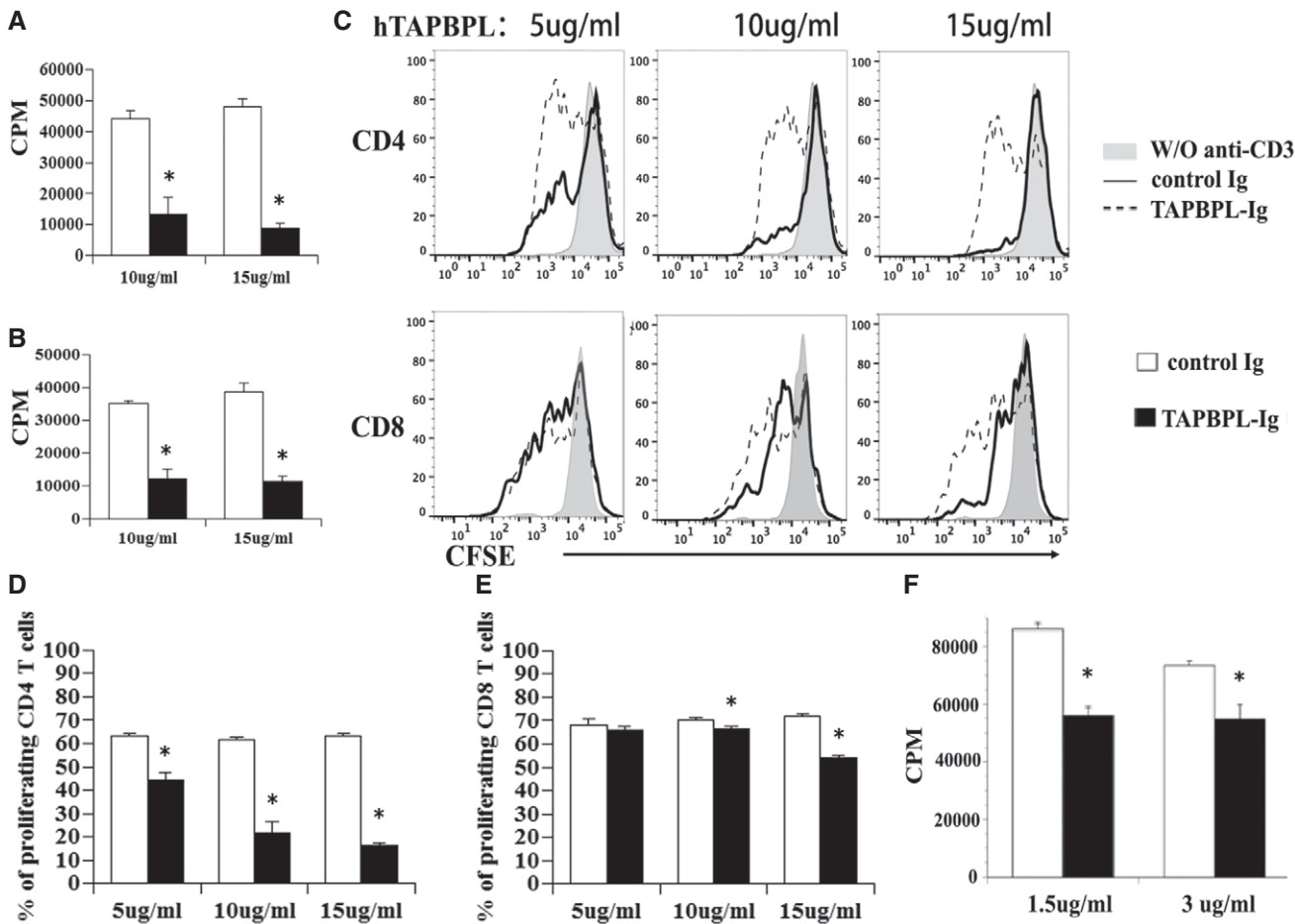

**Figure 5. The effect of hTAPBPL-Ig protein on T cell proliferation *in vitro*.**

A, B  T cells were purified from splenocytes of C57BL/6 mice by magnetic separation. The cells were cultured on plates pre-coated with (A) anti-CD3 antibody (1 µg/ml) or (B) anti-CD3 (1 µg/ml) and anti-CD28 (0.5 µg/ml) antibodies in the presence of graded doses of hTAPBPL-Ig (10 and 15 µg/ml) or equimolar amounts of control Ig (3.75 and 5.63 µg/ml) for 3 days. [³H] thymidine (1 µCi/well) was added to the cultures 12 h before harvest. T cell proliferation was measured by [³H] thymidine incorporation.

C–E  Splenic cells were labeled with CFSE and cultured in 96-well plates that were pre-coated with anti-CD3 antibody and hTAPBPL-Ig or control Ig for 3 days as in (A). The cells were analyzed for CFSE levels by CD4⁺ and CD8⁺ T cells. (C) Representative flow cytometric analysis of CFSE distribution of CD4⁺ and CD8⁺ T cells, and (D, E) statistical analysis of (D) CD4 and (E) CD8 T cell proliferation.

F  Purified human T cells were cultured with plate-bound anti-human CD3 antibody (1 µg/ml) in the presence of graded doses of hTAPBPL-Ig (1.5 and 3 µg/ml) or control Ig protein (1.5 and 3 µg/ml) for 3 days. Cell proliferation was measured by [³H] thymidine incorporation. The data are expressed as mean + SD ($n = 3$). Significance was calculated by two-tailed Student's *t*-test. *$P < 0.05$ compared with control Ig.

et al, 2012), we analyzed the activation of these molecules. As shown in Fig EV3A and B, PD-L1 inhibited the activation of AKT but not p38 and JNK, consistent with previous report (Patsoukis *et al*, 2012). In contrast, hTAPBPL-Ig inhibited the activation of AKT, p38, and JNK (Fig EV3A and B). The results suggest that the mechanisms by which TAPBPL and PD-L1 affect T-cell functions are different.

**Administration of hTAPBPL-Ig fusion protein ameliorates EAE in mice**

We next determined whether *in vivo* administration of hTAPBPL-Ig fusion protein could ameliorate EAE, a murine model of multiple sclerosis (MS). We first determined whether hTAPBPL-Ig could prevent EAE development. C57BL/6 mice were injected with MOG peptide to induce EAE. The mice were then injected with 25 µg hTAPBPL-Ig or control Ig protein on day 0 (the day that EAE was induced). EAE development was monitored over time. hTAPBPL-Ig significantly reduced the mean clinical scores throughout the entire 43-day time course (Appendix Fig S4A). At the end of the study, the spleens were harvested and analyzed for the percentages and activation of CD4⁺ and CD8⁺ T cells. hTAPBPL-Ig significantly deceased the percentage and number of CD4⁺ T cells and reduced the expression of CD69 by CD4⁺ and CD8⁺ T cells (Appendix Fig S4B–G). Meanwhile, the percentage and number of CD4⁺CD25⁺FoxP3⁺ Tregs were increased (Appendix Fig S4H and I). In addition, hTAPBPL-Ig decreased the percentages and numbers of CD4⁺ and CD8⁺ effector memory T cells but increased the percentages of naïve T cells (Appendix Fig S4J–M).

We then determined whether hTAPBPL-Ig could treat established EAE. C57BL/6 mice were induced to develop EAE as above. Once EAE symptoms occurred, the mice were injected with hTAPBPL-Ig or control Ig protein. As shown in Fig 6A, hTAPBPL-Ig significantly reduced the mean clinical scores. At the conclusion of the study, the spleen and spinal cord were harvested. A decreased proportion and number of CD4$^+$ T cells in the spleen of hTAPBPL-Ig-treated mice (Fig 6B and C) were observed. In contrast, hTAPBPL-Ig-treated mice had an increased percentage and number of CD4$^+$CD25$^+$FoxP3$^+$ Tregs (Fig 6D and E). hTAPBPL-Ig also reduced the expression of CD69 by CD4$^+$ and CD8$^+$ T cell, decreased the percentages and numbers of effector memory CD4$^+$ and CD8$^+$ T cells, but increased those of naïve T cells (Fig 6F–M). Furthermore, hTAPBPL-Ig-treated mice had decreased CNS-infiltrating CD4$^+$, Th1, and Th17 T cells (Fig 6N and O, and Fig EV4A), but increased the percentage and number of Tregs (Fig 6P and Q). However, the percentages of microglial cells, macrophages, and neutrophils in the CNS between hTAPBPL-Ig and control Ig-treated mice were not significantly different (Fig EV4B).

To determine whether Tregs play a role in the ameliorated EAE, we deleted Tregs by an anti-CD25 antibody and found that the deletion partly abolished the effect of hTAPBPL-Ig on EAE (Fig 6R and Fig EV4C). We then investigated whether hTAPBPL-Ig affects the differentiation of Tregs *in vitro*. As shown in Fig EV4D and E, hTAPBPL-Ig increased the differentiation of naïve T cells into Tregs, indicating that TAPBPL can directly affect Tregs.

We also analyzed MOG-specific T cell proliferation and cytokine production. When the splenocytes were stimulated with MOG *in vitro*, the proliferation of T cells, and IFNγ- or IL-17A-producing CD4$^+$ T cells from hTAPBPL-Ig-treated EAE mice were reduced (Fig 6S and Fig EV4F). Furthermore, we determined cytokine production from CD4$^+$ T cell in the draining lymph node of the EAE mice after stimulation with MOG *in vitro*. The production of IFNγ and IL-17A was also significantly decreased in hTAPBPL-Ig-treated EAE mice, whereas the production of IL-2 and GM-CSF was not significantly different (Fig EV4G).

Taken together, our results suggest that *in vivo* administration of hTAPBPL-Ig can prevent and treat MOG-induced EAE in mice. This amelioration of disease severity was associated with a decreased proportion of CD4$^+$ T cells and increased Tregs in the spleen and CNS, reduced activation of CD4$^+$ and CD8$^+$ T cells, and decreased Th1/Th17 cytokine production. In addition, hTAPBPL inhibited autoantigen-specific T cell proliferation and Th1/Th17 cytokine production.

**Anti-hTAPBPL mAb inhibits tumor growth *in vivo***

Since TAPBPL is highly expressed in tumor tissues and TAPBPL-Ig inhibits T-cell functions, we hypothesized that anti-TAPBPL antibody could block the inhibitory effect of TAPBPL, thereby enhancing antitumor immunity and inhibiting tumor growth *in vivo*. We first determined the ability of the anti-hTAPBPL mAb to neutralize the inhibitory activity of hTAPBPL on T cells *in vitro*. As shown in Fig 7A–C, the anti-hTAPBPL mAb (clone 54) neutralized the inhibitory activity of hTAPBPL on T cell proliferation and CD69 expression. However, the anti-hTAPBPL mAb alone did not affect the expression of CD69 and the proliferation of purified T cells (Fig EV5A and B).

We then assessed the ability of the mAb to treat cancer in mice injected s.c. with P388 leukemia cells. Anti-hTAPBPL mAb at 25 and 50 μg doses inhibited tumor growth in the model although the differences did not reach statistical significance for most time points; conversely, mAb at 100 μg dose significantly inhibited tumor growth (Fig 7D and Fig EV5C). At the conclusion of the study, we harvested the tumors and spleens to analyze immune cell populations. Anti-hTAPBPL mAb treatment resulted in increased percentages of tumor-infiltrating CD4$^+$ and CD8$^+$ T cells (Fig 7E–H and Fig EV5D), but decreased percentage of CD4$^+$CD25$^+$FoxP3$^+$ Tregs (Fig 7I and J), as determined by flow cytometry. The increased tumor-infiltrating CD4$^+$ and CD8$^+$ T cells were confirmed by immunohistochemistry (Fig 7K and L, and Fig EV5E). Analysis of splenic T-cell populations showed that anti-hTAPBPL mAb-treated spleens also had increased percentages of CD4$^+$ and CD8$^+$ T cells, but a decreased percentage of CD4$^+$CD25$^+$FoxP3$^+$ Tregs (Fig EV5F–H).

To determine whether CD8 T cells mediated the antitumor activity, groups of the P388 cancer-bearing mice were also injected with anti-CD8 antibody. Depletion of CD8 T cells partly abrogated the antitumor effect of the anti-hTAPBPL mAb (Fig EV5I), suggesting that CD8 T cells might partly mediate the antitumor activity of the mAb.

To confirm the tumor suppression effect, we also used a B16F10 melanoma murine model. Although B16F10 cells express low level of TAPBPL, anti-hTAPBPL mAb also significantly inhibited the growth of B16F10 melanoma cells, as compared to control antibody treatment (Fig 7M and Fig EV5J). We did not observe any toxicity with the anti-hTAPBPL mAb treatment.

Together, our results suggest that the anti-TAPBPL mAb can inhibit tumor growth *in vivo*, which is related to neutralizing the inhibitory activity of TAPBPL on T cells in the tumor and spleen.

## Discussion

The present study describes the identification of TAPBPL as a novel T cell co-inhibitory molecule. TAPBPL was originally identified on chromosome position 12p13.3, a region somewhat paralogous to the MHC (Du Pasquier, 2000; Teng *et al*, 2002; Hermann *et al*, 2015; Morozov *et al*, 2016). TAPASIN, encoded by the TAP binding protein (TAPBP) gene, links MHC I molecules to the transporter associated with TAP. Although TAPBPL shares TAPASIN's ability to bind to MHC class I, TAPBPL is not simply a TAPASIN analogue (Hermann *et al*, 2015). For example, TAPASIN has an ER retention motif, while TAPBPL does not. We found that TAPBPL has the characteristics of B7 family molecules. For example, TAPBPL shares a significant sequence similarity with some of the B7 family members. The extracellular region of TAPBPL contains an IgV and an IgC domain. Most importantly, TAPBPL-Ig protein inhibits the proliferation and activation of T cells. Therefore, TAPBPL is a new member of the B7 family or B7 family-related molecule.

We have shown that TAPBPL protein is expressed on the surface of APCs, including DCs, monocytes, macrophages, and B cells, and that the expression on DCs and monocytes was upregulated upon activation. Like other B7 family molecules, the expression of TAPBPL protein on APCs likely plays an important role in regulating T-cell functions. In addition, TAPBPL is expressed on T cells, which may also act in cis to T cells. Our results are largely consistent with

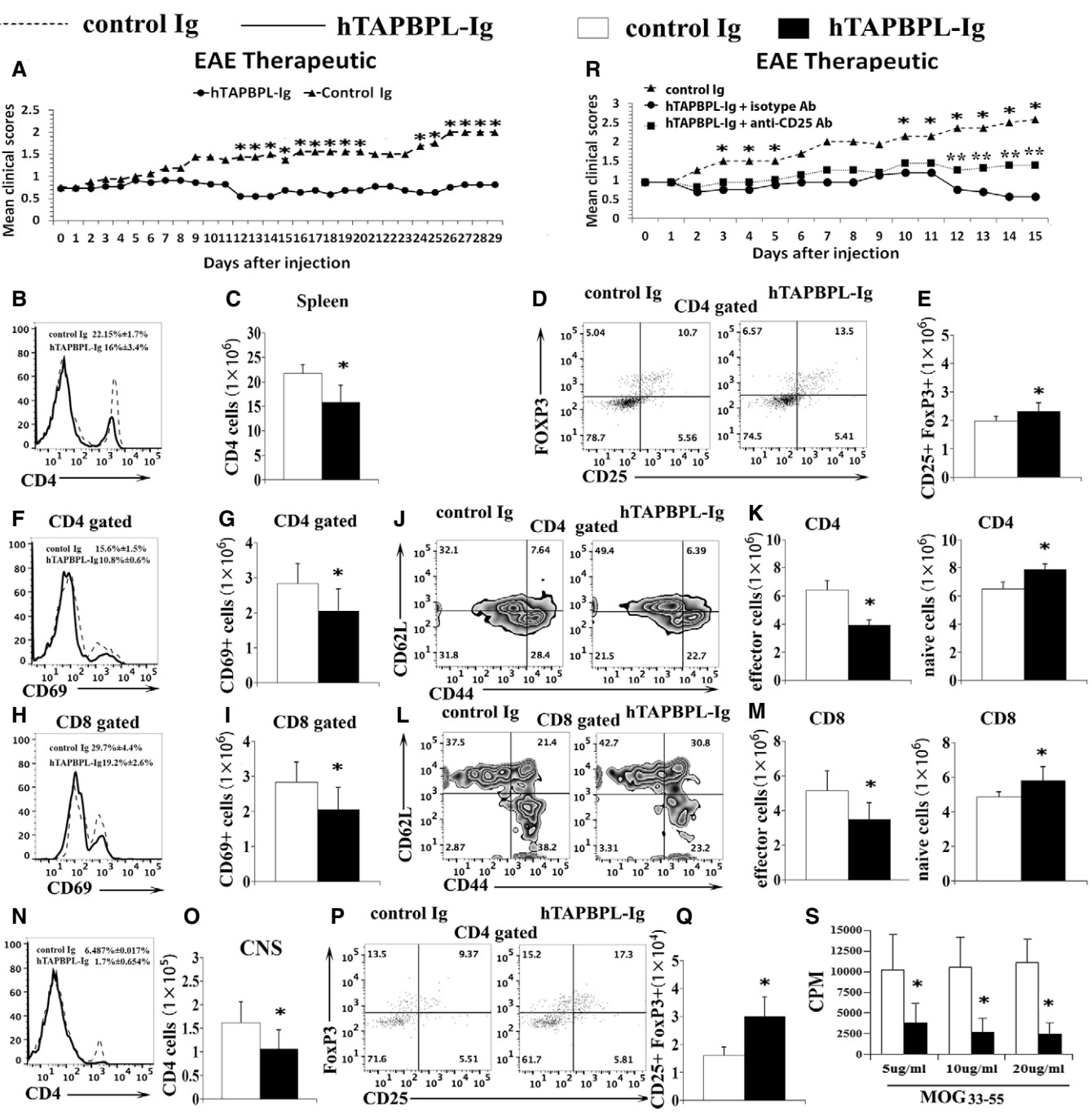

**Figure 6. hTAPBPL-Ig ameliorates established EAE in mice.**

C57BL/6 mice were immunized with 200 μg MOG35-55 emulsified in CFA and 500 ng of purified Bordetella pertussis toxin. When EAE symptoms occurred, the mice were injected i.p. with 25 μg hTAPBPL-Ig or control Ig protein 3 times per week.

A–S   (A) Mean clinical scores. (B–Q, S) At the end of the studies, spleens and spinal cords were harvested and analyzed for the (B, D, F, H) percentages and (C, E, G, I) numbers of (B and C) CD4[+] T cells and (D and E) CD4[+]CD25[+]FoxP3[+] Tregs, the expression of CD69 by (F and G) CD4[+] and (H and I) CD8[+] T cells, the (J, L) percentages and (K, M) number of CD44[hi] CD62L[lo] effector memory and CD44[lo] CD62L[hi] naïve (J and K) CD4[+] and (L and M) CD8[+] T cells in the spleens. The (N, P) percentages and (O, Q) numbers of CNS-infiltrating (N, O) CD4[+] T cells and (P, Q) CD4[+]CD25[+]FoxP3[+] Tregs. (R) Deletion of Tregs partly reverses the effect of TAPBPL on EAE. Groups of EAE mice as in (A) were injected i.p. with anti-CD25 antibody. Mean clinical scores are shown. (S) Splenocytes from (B) were cultured with graded doses of MOG35-55 for 72 h. [3H] thymidine (1 μCi/well) was added to the cultures 12 h before harvest. T cell proliferation was measured by [3H] thymidine incorporation. The data are expressed as mean + SD and representative of 3 independent experiments with $n = 8$/group. Significance was calculated by two-tailed Student's t-test except (R) that was analyzed by two-way ANOVA with Tukey test. *$P < 0.05$, hTAPBPL-Ig group was compared with control Ig group; **$P < 0.05$, hTAPBPL-Ig + anti-CD25 antibody group was compared with hTAPBPL-Ig + isotype antibody group.

the data in the BioGPS database showing that the TAPBPL gene is expressed in DCs, monocytes, macrophages, and B cells. It has been reported that TAPBPL mRNA is expressed in lymphoid organs including the thymus and spleen and that TAPBPL protein is expressed on the cell surface (Teng et al, 2002). Our results are in agreement with that report.

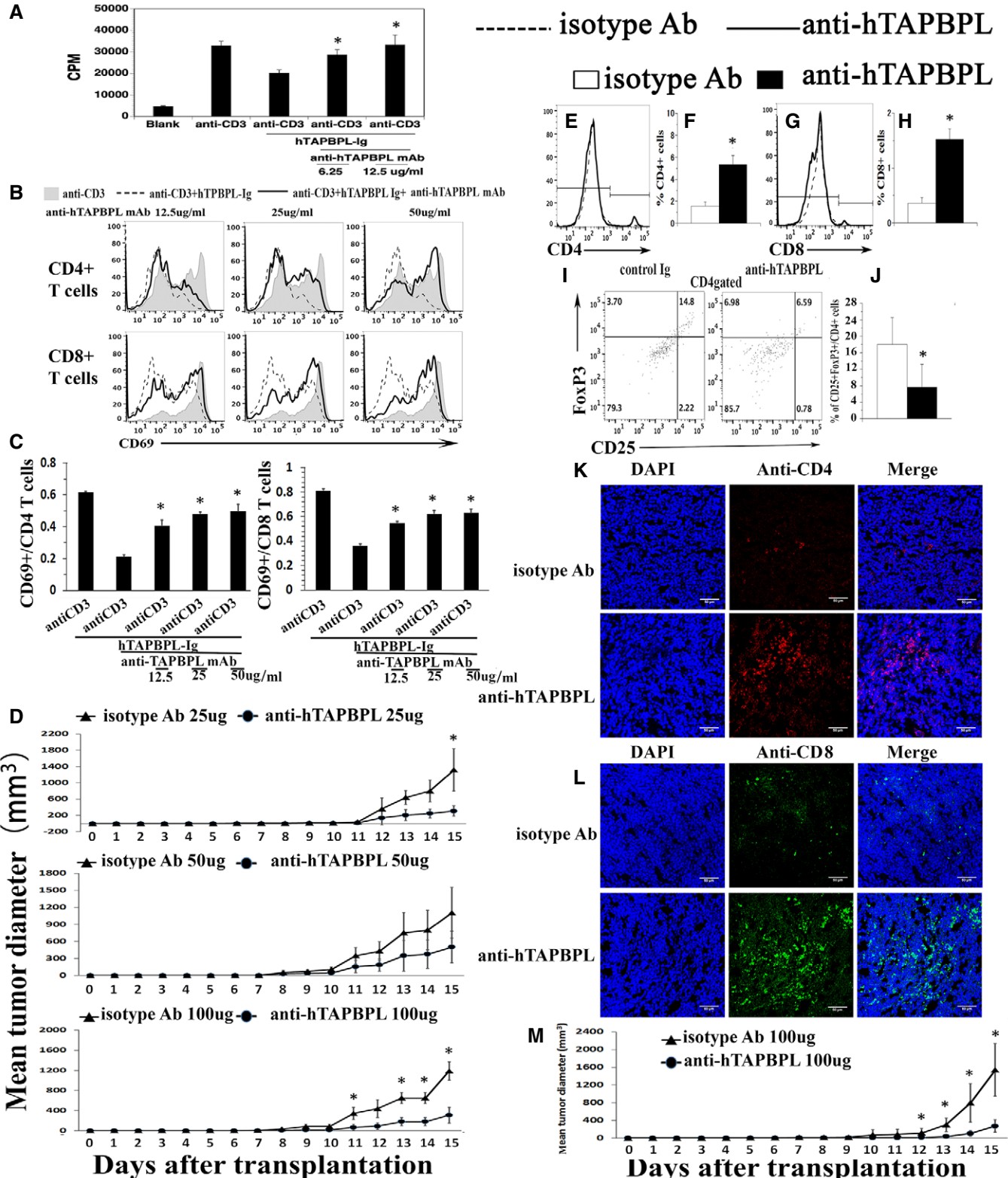

Figure 7.

◀

**Figure 7. An anti-hTAPBPL mAb neutralizes T cell inhibitory activity of hTAPBPL *in vitro* and inhibits tumor growth *in vivo*.**

A–C  Anti-hTAPBPL mAb (clone 54) neutralized the inhibitory activity of hTAPBPL-Ig on (A) T cell proliferation, and (B, C) CD69 expression by CD4$^+$ and CD8$^+$ T cells *in vitro*. (D–L) DBA/2J mice were injected s.c. with $1 \times 10^5$ P388 murine leukemia cells, followed by injection of the anti-hTAPBPL mAb (25, 50, or 100 μg) or isotype Ab (25, 50, or 100 μg) 3 times per week.

D    Tumor size was measured over time. The mean tumor diameter (mm$^3$) $\pm$ SD at the indicated time points is shown.

E–J  At the end of the studies, tumors from 100 μg anti-hTAPBPL mAb- or isotype Ab-treated mice were harvested. Single-cell suspension of the tumors was analyzed for the percentage of (E, F) CD4$^+$ T cells, (G, H) CD8$^+$ T cells, and (I and J) CD4$^+$CD25$^+$FoxP3$^+$ Tregs by flow cytometry.

K, L  Tumor-infiltrating CD4$^+$ and CD8$^+$ T cells were examined by immunofluorescence.

M    C57BL/6 mice were injected s.c. with $1 \times 10^5$ B16F10 melanoma cells and intratumorally with anti-hTAPBPL mAb or isotype Ab (100 μg) 3 times per week. Tumor size was measured over time. The data are expressed as mean $\pm$ SD and representative of two independent experiments with similar results ($n = 6$/group/time). Significance in (A, C) was calculated by one-way ANOVA with Dunnett test and others by two-tailed Student's *t*-test. *$P < 0.05$, anti-hTAPBPL Ab group was compared with isotype Ab group.

We have demonstrated that *in vivo* administration of hTAPBPL-Ig attenuates autoimmune disease EAE. Studies have shown that EAE is mainly mediated by CD4$^+$ T cells. Although hTAPBPL-Ig treatment resulted in a decreased proportion of CD4$^+$ T cells, it did not significantly reduce the percentages of CD8$^+$ T cells. This is not consistent with the *in vitro* data and is probably related to the EAE model. However, hTAPBPL-Ig treatment reduced the activation of CD4$^+$ and CD8$^+$ T cells in the EAE model, which is consistent with the *in vitro* data. Most importantly, hTAPBPL inhibited antigen-specific T cell proliferation and Th1/Th17 cytokine production in the EAE mice. We have also shown that hTAPBPL-Ig increases the percentage and/or number of Tregs in the spleen and the CNS. Deletion of Tregs partly abolishes the inhibition of EAE by hTAPBPL-Ig, suggesting that Tregs may share a role in the effect of TAPBPL on EAE. Our data also suggest that TAPBPL-Ig can enhance the differentiation of naïve T cells into Tregs. It remains to be determined whether TAPBPL affects thymic Treg development.

We also found that TAPBPL protein was highly expressed on some tumor tissues, which is consistent with the data in the Human Protein Atlas database (https://www.proteinatlas.org). A significant increase in TAPBPL expression was also observed in quantitative trait locus (eQTL) studies of various tumors (Chen *et al*, 2014). Furthermore, high expression of TAPBPL was associated with a poor outcome (Chen *et al*, 2014). Our results and those from others suggest that, like PD-L1 and other T cell inhibitory molecules, TAPBPL may be involved in immune evasion of cancer. In addition, it has been reported that TAPBPL over-expression in HeLa cells reduces the cell surface expression of MHC class I, resembling the phenotype of a TAPASIN-deficient cell (Hermann *et al*, 2015). The reduced cell surface expression of MHC class I could also inhibit antitumor immunity. With several blocking antibodies for PD-L1/PD-1 and CTLA-4 approved by FDA, targeting T cell inhibitory molecules has become one of the most successful and important strategies for treating cancer patients. We have shown that an anti-hTAPBPL mAb significantly inhibits tumor growth *in vivo*, with a concomitant increase in tumor-infiltrating CD4$^+$ and CD8$^+$ T cells and a decrease in the percentage of Tregs. Since hTAPBPL protein is expressed on human cancer cells and APCs, the anti-hTAPBPL mAb has the potential to be used in the treatment of cancer patients by blocking the inhibitory activity of hTAPBPL.

In summary, we describe TAPBPL as a novel T cell co-inhibitory molecule. TAPBPL protein is expressed on APCs and in tumor tissues. TAPBPL-Ig fusion protein inhibits T cell proliferation and activation *in vitro*. *In vivo* administration of TAPBPL-Ig protein attenuates EAE. Anti-TAPBPL antibody inhibits tumor growth in mouse models. Therefore, targeting the TAPBPL has the potential to

be used in the treatment of autoimmune diseases (such as MS) and transplant rejection, as well as cancer and infection.

# Materials and Methods

### Bioinformatics analysis of TAPBPL

Sequence alignments of the extracellular domains of human TAPBPL and existing B7 family members, as well as the full sequences of human and mouse TAPBPL proteins, were analyzed via the Clustal W program in MacVector 16.0.5 (MacVector, Inc.). The leader peptide, transmembrane, and Ig-like domain were predicted with SignalP 4.0 (http://www.cbs.dtu.dk/services/SignalP), TMHMM server version 2.0 (http://www.cbs.dtu.dk/services/TMHMM/), and InterPro (https://www.ebi.ac.uk/interpro).

### Cloning and purification of TAPBPL

The extracellular domain of hTAPBPL or mTAPBPL was cloned and fused into a pCMV6-AC-FC-S expression vector containing the constant region of mouse IgG2a (ORIGENE, Rockville, MD). The vector was transfected into HEK-293 cells. The fusion proteins were purified from the supernatant using Protein G Sepharose 4 Fast Flow according to the manufacturer's instructions (GE Healthcare). Purified proteins were verified by SDS–PAGE, Coomassie staining, and Western blot. Protein concentration was quantified using the Pierce™ BCA Protein Assay Kit (Pierce, Rockford, IL). Control Ig (recombinant mouse IgG2a Fc protein) was purchased from BXCell (West Lebanon, NH).

### Mice

C57BL/6 and DBA/2J mice were purchased from Jackson Laboratory. The mice were used in accordance with protocols approved by the Institutional Animal Care and Use Committee of the University of Connecticut.

### Flow Cytometry analysis

Single-cell suspensions of organs and tumors were stained with the fluorochrome-conjugated antibodies protein as described (Cui et al, 2018; Su *et al*, 2019; Tian *et al*, 2019). For intracellular staining, the cells were first permeabilized with a BD Cytofix/Cytoperm solution for 45 min at 4°C. Direct or indirect staining of fluorochrome-conjugated antibodies included CD4 (1:100), CD8 (1:100), CD19 (1:100),

  

B220 (1:100), CD11c (1:60), CD11b (1:100), F4/80 (1:60), CD44 (1:60), CD62L (1:60), CD69 (1:60), CTLA-4 (1:40), CD28 (1:40), PD-1 (1:40), BTLA (1:40), and ICOS (1:40) (BioLegend, or BD Biosciences, San Jose, CA, San Diego, CA). Anti-TAPBPL monoclonal antibodies were generated from our laboratory. Biotinylation of TAPBPL-Ig and control Ig proteins was performed by EZ-Link™ Sulfo-NHS-LC-Biotin kit (Thermo Scientific TE260201). The samples were analyzed on a LSRFortessa X-20 Cell Analyzer (BD Biosciences). Data analysis was done using FlowJo software (Ashland, OR).

### Histopathology

Human Multiple Normal and Tumor Tissue Arrays were purchased from BioChain (Newark, CA). The sections were subjected to antigen unmasking and incubated with anti-TAPBPL mAb (1:100), followed by ImmPRESS VR Polymer HRP anti-mouse IgG reagent. The sections were then developed with the ImmPACT DAB Peroxidase Substrate and counterstained with hematoxylin (Vector Laboratories) according to the manufacturer's instructions.

### *In vitro* T-cell assays

Normal human peripheral blood $CD3^+$ Pan T cells that were negatively isolated from mononuclear cells using an indirect immunomagnetic Pan-T labeling system were purchased from ALLCELLS, LLC (Alameda, CA). Murine $CD3^+$ T cells were purified from C57BL/6 mice by an immunomagnetic system (Miltenyi, Auburn, CA), and the purity of the cells was usually > 95%. T cells were stimulated with anti-CD3 antibody, or anti-CD3 plus anti-CD28 antibodies (BioLegend) in the presence of TAPBPL-Ig or control Ig for 3 days. Proliferative response was assessed by pulsing the culture with 1 μCi of [³H] thymidine (PerkinElmer, Inc., Downers Grove, IL) 12 h before harvest. Incorporation of [³H] thymidine was measured by liquid scintillation spectroscopy (PerkinElmer, Inc.). For CFSE assay, splenocytes were labeled with CFSE (Thermo Fisher Scientific) and stimulated with anti-CD3 antibody in the presence of TAPBPL-Ig or control Ig. The cells were analyzed by flow cytometry.

### Induction and assessment of EAE

Mouse $MOG_{35-55}$ (GL Biochem, Shanghai, China) was emulsified in complete Freud's adjuvant (Sigma-Aldrich, St Louis, MO, USA) supplemented with mycobacterium tuberculosis H37Ra (Difco Laboratories, Detroit, MI). Mice were injected s.c. with the MOG at 4 points in the dorsal flank on day 0. The mice were also injected i.p. with 500 ng of purified Bordetella pertussis toxin (Sigma-Aldrich). The mice were injected i.p. with hTAPBPL-Ig, or control Ig, and observed for clinical scores based on the following scale: 0, normal; 0.5, partially limp tail; 1, paralyzed tail; 2, loss in coordinated movement, hind limb paresis; 2.5, one hind limb paralyzed; 3, both hind limbs paralyzed; 3.5, hind limbs paralyzed, weakness in forelimbs; 4, forelimbs paralyzed; and 5, moribund or dead. As required by animal ethics, mice were euthanized beyond a clinical score of 4. To delete Tregs in EAE mice, once EAE symptoms occurred, the mice were injected i.p. with 250 μg anti-CD25 antibody (clone PC61, from BioXCell) or isotype antibody on days 0 and +4.

### The paper explained

#### Problem

T cells are regulated by stimulatory and inhibitory molecules. Among the T-cell regulators, the B7 family is of central importance. Several drugs that target B7 family T cell inhibitory molecules, such as PD-L1/PD-1 and CTLA-4, have been used in the treatment of cancer or autoimmune disease. However, complete and durable responses are only seen in a fraction of treated patients.

#### Results

We identify a novel T cell inhibitory molecule TAPBPL that shares sequence and structural homology with existing B7 family members. TAPBPL protein is expressed on APCs and cancer cells. The TAPBPL receptor is expressed on activated CD4 and CD8 T cells. Recombinant hTAPBPL-Ig protein inhibits the proliferation and activation of T cells *in vitro*. Administration of hTAPBPL-Ig protein attenuates experimental autoimmune encephalomyelitis (EAE) in mice. Furthermore, an anti-TAPBPL monoclonal antibody neutralizes the inhibitory activity of hTAPBPL-Ig on T cells, enhances antitumor immunity, and inhibits tumor growth in animal models.

#### Impact

Our results suggest that therapeutic intervention of the TAPBPL inhibitory pathway may represent a new strategy to treat cancer and autoimmune diseases. Targeting this inhibitory pathway may be effective in the treatment of the subtypes of patients who are resistant to PD-L1/PD-1 and CTLA-4 targeting. Furthermore, combination immunotherapy that targets the TAPBPL and PD-L1/PD-1 (and/or CTLA-4) inhibitory pathways may provide a promising approach to facilitate rapid and complete responses for patients with autoimmune disease or cancer.

### Generation of hTAPBPL monoclonal Antibodies (mAbs)

BALB/c mice were immunized with 50 μg hTAPBPL-Ig protein emulsified in complete Freund's adjuvant (CFA) on day 0 and boosted on day 14 and day 21 in the same protein quantity in incomplete Freund's adjuvant (IFA). The mice were boosted with 50 μg hTAPBPL-Ig without IFA 3 times (days 28, 29, and 30). On day 31, the spleens were harvested from the immunized mice. Single-cell suspension of the splenocytes was fused to X63-Ag8.653 myeloma cells to produce hybridomas. ELISA was performed to identify the hybridomas that could produce anti-hTAPBPL mAbs reacting with hTAPBPL, but not with control Ig protein. The anti-hTAPBPL mAbs were further screened for the ability to neutralize the inhibitory activity of hTAPBPL-Ig on T cell proliferation and activation. Anti-hTAPBPL mAb was purified from the supernatant of a hybridoma line using Protein G Sepharose 4 Fast Flow according to the manufacturer's instructions (GE Healthcare).

### Evaluation of local tumor growth

Murine P388 leukemia cells and B16F10 melanoma cells were obtained from ATCC. The P388 or B16F10 cancer cells ($1 \times 10^5$) were injected s.c. into syngeneic DBA/2J or C57BL/6 mice, respectively. Anti-hTAPBPL mAb or isotype Ab was then injected into the tumor injection site. Tumor size (volume) was determined by caliper measurements of the shortest (A) and longest (B) diameter, using the formula $V = (A^2B)/2$. For *in vivo* cell depletion of CD8 T cells, mice were injected i.p. with 500 μg anti-CD8 antibody

(clone 2.43, from BioXCell) on days −3, −1, and +4 of the cancer cell injection.

## Confocal microscopy

Frozen sections of tumor tissue were prepared as described previously (Jin *et al*, 2011; Lai *et al,* 2012; Lai *et al,* 2013; Song *et al*, 2015). The slides were incubated with purified rat anti-mouse CD4 or CD8 antibody (1:100) (BioLegend) followed by Alexa Fluor 546-labeled goat anti-Rat IgG (H + L) (1:200) or Alexa Fluor 488-labeled chicken anti-Rat IgG (H + L) (1:200) (Invitrogen). The slides were mounted with mounting medium with DAPI (VECTASHIELD HardSet^TM H1500) and observed under a Nikon A1R confocal laser scanning microscope.

## Statistical analysis

For comparing means of 2 groups, two-tailed Student's *t*-test was used. For comparing means of multiple groups, significance was determined using one-way ANOVA with Dunnett test or two-way ANOVA with Tukey test. Differences with $P < 0.05$ were considered statistically significant. The exact *p*-values are listed in Appendix Table S1. Data were represented as mean ± SD. For animal experiments, mice were randomly allocated into control or experimental groups. All the samples or animals were analyzed in blind when possible.

# Data availability

This study includes no data deposited in external repositories.

**Expanded View** for this article is available online.

## Acknowledgments

We would like to thank Qingquan Chen for his technical help. This work was supported by grants from NIH (1R01AI123131) and Connecticut Regenerative Medicine Research Fund (16-RMB-UCONN-02).

## Author contributions

LL designed the experiments. YL, CC, MS, HL, JZ, LH, YH, DX, XW, and LL performed the experiments. YL, CC, MS, LKS, QD, and LL analyzed the data. LL and YL wrote the draft. LKS provided major editorial comments. All authors participated in editing the final version of the manuscript.

## Conflict of interest

The authors declare that they have no conflict of interest.

## For more information

i  https://www.proteinatlas.org; http://www.cbs.dtu.dk/services/SignalP.
ii  http://www.cbs.dtu.dk/services/TMHMM/; https://www.ebi.ac.uk/interpro.

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
