## [Review Process File · EMBO Molecular Medicine]

Identification of TAPBPL as a novel negative regulator of T cell function

Yujun Lin, Cheng Cui, Min Su, Lawrence K. Silbart, Haiyan Liu, Jin Zhao, Lang He, Yuanmao Huang, Dexin Xu, Xiaodan Wei, Qian Du and Lijun Lai

DOI: [10.15252/emmm.202013404](https://doi.org/10.15252/emmm.202013404)

Corresponding author: Lijun Lai (lijun.lai@uconn.edu)

Review Timeline:

Submission Date:	4th Sep 20
Editorial Decision:	24th Sep 20
Revision Received:	15th Feb 21
Editorial Decision:	5th Mar 21
Revision Received:	14th Mar 21
Accepted:	16th Mar 21

Editor: Zeljko Durdevic

Transaction Report:

24th Sep 2020

Dear Dr. Lai,

Thank you for the submission of your manuscript to EMBO Molecular Medicine. We have now received feedback from the three reviewers who agreed to evaluate your manuscript. As you will see from the reports below, the referees acknowledge the interest and novelty of the study but also raise serious and partially overlapping concerns that should be addressed in a major revision. Particular attention should be given to experiments that would strengthen the translational aspect of the study.

Addressing the reviewers' concerns in full will be necessary for further considering the manuscript in our journal, and acceptance of the manuscript will entail a second round of review. EMBO Molecular Medicine encourages a single round of revision only and therefore, acceptance or rejection of the manuscript will depend on the completeness of your responses included in the next, final version of the manuscript. For this reason, and to save you from any frustrations in the end, I would strongly advise against returning an incomplete revision.

Considering the extent of the revision, I am happy to extend the revisions time to 6 months. We realize that the current situation is exceptional on the account of the COVID-19/SARS-CoV-2 pandemic. Therefore, please let us know if you need more than six months to revise the manuscript.

I look forward to receiving your revised manuscript.

Yours sincerely,

Zeljko Durdevic

***** Reviewer's comments *****

Referee #1 (Remarks for Author):

In this paper, the authors report a study targeting TAPBPL for autoimmunity and tumor immunity. In this study, they have identified a new B7 family member called TAPBPL. They showed its expression in lymphoid cells, myeloid cells, normal tissues and cancer tissues. In addition, they demonstrated that the recombinant TAPBL inhibited T cell activation and proliferation in vitro and manifestation of EAE in vivo. In addition, they showed that a TAPBPL antibody slowed tumor growth. The finding is quite novel. The data are largely supportive of their conclusion. There are

some concerns with some of the data being generated without proper control. Antitumor efficacy should be more rigorously examined with multiple tumor models.

Major critics:

Fig2. Since the antibody is generated in this lab and this is the first time it is reported, proper control should be provided to prove the specificity of the antibody. For instance, a cell line that expresses this protein should be used as a positive control and a cell line in which this gene has been deleted should be used as negative control. The specificity for IHC staining should also be demonstrated using similar controls. In addition, the affinity and specificity of this antibody should be shown. As for the level of expression, RT-QPCR data should also be presented.

Fig6. The underlying cellular mechanism by which hTAPBPL-Ig regulates T cell responses during EAE has not been fully examined. The frequency of Th17 and Th1 cells in the CNS should be shown. Likewise, the percentage of microglial cells, macrophages and neutrophils in the CNS should be shown. In addition, the frequency of MOG peptide-specific Th17 and Th1 cells in the spleen should be analyzed.

Fig7. It is assumed, based on the in vivo experiments, that the hTAPBPL mAb cross-reacts with mouse TAPBPL. However, these data need to be shown with proper positive and negative control. Does this antibody alone increase T cell proliferation in vitro? The infiltration of T cells in panels K and L needs to be quantified. It is generally more desirable to use at least two tumor models to demonstrate the efficacy of this reagent. What is the antitumor efficacy of the hTAPBPL mAb for B16 and MC38 models? In addition, does combination of hTAPBPL mAbs and PD-1 mAbs produce greater antitumor efficacy? Does the hTAPBPL mAbs treatment result in any toxicity? The body weight of these mice should be shown.

Minor problems

Fig 6 The Y axis labels for many figures are too small.

Fig 7 The labels in B and C are very hard to read. The gating strategy for TIL should be shown.

Referee #2 (Remarks for Author):

In this manuscript, Lin and colleagues report the identification of TAPBPL. They propose that TAPBPL is a novel T cell co-inhibitory molecule. To support their claims, the authors present data showing that TAPBPL can be expressed on different immune and tumor cells and, importantly, that manipulation of TAPBPL signaling in vivo in EAE and cancer can be performed for therapeutic purposes. Given the importance of co-inhibitory receptors, notably in the cancer field, this study could be relevant in the context of cancer treatment. There are however issues with the study presented that limit the translational potential of the work. My concerns are as follows:

- 1) In places, the doses of stimulating agents used to study TAPBPL expression are very high. This applies for instance on the doses of IFN γ used (Supplementary Figure 1) and also, even more importantly, on the doses of LPS used. It is curious why the authors need to use such high doses to achieve expression. Showing the expression levels of control molecules (for instance co-stimulation molecules on DC) may help to clarify the relevance of the proposed biological effects.

- 2) It is not clear how statistics were computed in the whole manuscript.
- 3) The effects on T cell proliferation are shown and convincing. However, the mechanism of action of TAPBPL signaling remains elusive. Effects on T cell activation should be explored. What about cytokine release?
- 4) It remains difficult to understand from in vivo experiments how signaling induced by TAPBPL works. What are the cells accounting for the observed effects in EAE? In cancer? The authors discuss that Tregs may be involved but in the context of this study, this needs to be investigated in more depth.

Overall, the study has potential medical relevance but remains unfortunately preliminary in its current form.

Referee #3 (Remarks for Author):

The authors reported a novel molecule TAPBPL which showed inhibitory function on T cell activation. The study was well logically designed, the data are overall convincing. Some technical issues need to be addressed.

1. The expression of TAPBPL on various immune cells seem not related to their activation status, which is different from many B7 family members. This should be validated with other methods, such as RT-PCR. Can type I or II IFN stimulation upregulate TAPBPL on dendritic cells or macrophages?
2. Staining of TAPBPL on tumor tissues are not clear. What substrate was used for TAPBPL visualization? How is the section counterstained? In addition, additional methods, such as RT-PCR, are suggested to confirm the results.
3. Figure 4 and Figure 5 can be inversed, since T cell activation is before T cell proliferation.
4. The increased Treg population in EAE model upon hTAPBPL-Ig treatment and reduced Treg in tumor model upon anti-hTAPBPL treatment are interesting. Is this a direct or indirect effect of hTAPBPL manipulation? Can hTAPBPL-Ig increase Treg differentiation in vitro?
5. The authors confirmed the neutralizing role of anti-hTAPBPL on hTAPBPL-Ig's inhibitory effect. This is not surprising. Can anti-hTAPBPL increase T cell activation/proliferation? This would be very informative to be shown at least using in vitro assay.
6. The authors use human TAPBPL related reagents for mouse study. Can mTAPBPL-Ig or anti-mTAPBPL demonstrate more clear-cut results? This is highly expected.

Comments from Editor: Thank you for the submission of your manuscript to EMBO Molecular Medicine. We have now received feedback from the three reviewers who agreed to evaluate your manuscript. As you will see from the reports below, the referees acknowledge the interest and novelty of the study but also raise serious and partially overlapping concerns that should be addressed in a major revision. Particular attention should be given to experiments that would strengthen the translational aspect of the study.

Addressing the reviewers' concerns in full will be necessary for further considering the manuscript in our journal, and acceptance of the manuscript will entail a second round of review. EMBO Molecular Medicine encourages a single round of revision only and therefore, acceptance or rejection of the manuscript will depend on the completeness of your responses included in the next, final version of the manuscript. For this reason, and to save you from any frustrations in the end, I would strongly advise against returning an incomplete revision.

Response 1: We sincerely thank the editor for the very thoughtful judgment of the reviewers' comments. We have attempted to respond to all their comments, as indicated below and in the revised manuscript (yellow highlighting). We hope that these revisions are satisfactory.

Review #1:

General Comment: In this paper, the authors report a study targeting TAPBPL for autoimmunity and tumor immunity. In this study, they have identified a new B7 family member called TAPBPL. They showed its expression in lymphoid cells, myeloid cells, normal tissues and cancer tissues. In addition, they demonstrated that the recombinant TAPBPL inhibited T cell activation and proliferation in vitro and manifestation of EAE in vivo. In addition, they showed that a TAPBPL antibody slowed tumor growth. The finding is quite novel. The data are largely supportive of their conclusion. There are some concerns with some of the data being generated without proper control. Antitumor efficacy should be more rigorously examined with multiple tumor models.

Response to General Comment: We thank the reviewer for his/her thoughtful and positive evaluation of our study and the comments that have improved the manuscript.

Comment 1: Fig2. Since the antibody is generated in this lab and this is the first time it is reported, proper control should be provided to prove the specificity of the antibody. For instance, a cell line that expresses this protein should be used as a positive control and a cell line in which this gene has been deleted should be used as negative control. The specificity for IHC staining should also be demonstrated using similar controls. In addition, the affinity and specificity of this antibody should be shown. As for the level of expression, RT-QPCR data should also be presented.

Response 1: We have provided new data showing that the anti-TAPBPL mAb reacted with hTAPBPL-Ig and mTAPBPL, but not control Ig protein (Appendix Fig S1B). Furthermore, the anti-hTAPBPL mAb stained parent P388 leukemia cells, but not mTAPBPL siRNA treated P388 cells (Appendix Fig S1C). In addition, the anti-TAPBPL mAb neutralized the inhibitory activity of hTAPBPL on T cell proliferation and CD69 expression (Fig 7A-C). We have used isotype Ab as a negative control for the IHC staining (Figure 2C, D). We have also provided new data

showing the expression of the TAPBPL mRNA in the immune cells and cancer cells examined by qRT-PCR (Fig EV1B, F).

Comment 2: Fig6. The underlying cellular mechanism by which hTAPBPL-Ig regulates T cell responses during EAE has not been fully examined. The frequency of Th17 and Th1 cells in the CNS should be shown. Likewise, the percentage of microglial cells, macrophages and neutrophils in the CNS should be shown. In addition, the frequency of MOG peptide-specific Th17 and Th1 cells in the spleen should be analyzed.

Response 2: We have provided new data showing that hTAPBPL-Ig-treated EAE mice had decreased CNS-infiltrating Th1 and Th17 T cells (Fig EV4A). We have also shown the data for the percentages of microglial cells, macrophages and neutrophils in the CNS between hTAPBPL-Ig and control Ig-treated mice (Fig EV4B). In addition, we have provided new data showing that the percentage of MOG peptide-specific Th1 and Th17 cells in the spleen and the draining lymph nodes of hTAPBPL-Ig-treated mice was reduced (Fig EV4F, G).

Comment 3: Fig7. It is assumed, based on the *in vivo* experiments, that the hTAPBPL mAb cross-reacts with mouse TAPBPL. However, these data need to be shown with proper positive and negative control. Does this antibody alone increase T cell proliferation *in vitro*? The infiltration of T cells in panels K and L needs to be quantified. It is generally more desirable to use at least two tumor models to demonstrate the efficacy of this reagent. What is the antitumor efficacy of the hTAPBPL mAb for B16 and MC38 models? In addition, does combination of hTAPBPL mAbs and PD-1 mAbs produce greater antitumor efficacy? Does the hTAPBPL mAbs treatment result in any toxicity? The body weight of these mice should be shown.

Response 3: As indicated in our response to Comment 1, we have shown that the anti-hTAPBPL mAb cross-reacted with mouse and human TAPBPL (Appendix Fig S1B) and stained parent mouse P388 leukemia cells, but not mTAPBPL siRNA treated P388 cells (Appendix Fig S1C). In addition, this mAb neutralized the inhibitory activity of hTAPBPL on T cell proliferation and CD69 expression (Fig 7A-C). This mAb alone did not significantly increase the activation and proliferation of T cells *in vitro* probably because the cultures contained purified T cells only (but not APCs that express TAPBPL) (Fig EV5A, B). The infiltration of T cells in panels K and L has been quantified (Fig EV5E). In addition, we have provided new data showing the anti-hTAPBPL mAb inhibits tumor growth in the B16 melanoma model (Fig 7M). Because of limited time allowed for the revision, we will perform further studies to determine whether the combination of anti-hTAPBPL mAbs and PD-1 mAbs can produce greater antitumor efficacy. We did not observe any toxicity with the anti-hTAPBPL mAb treatment (line 345). The body weights of these mice have been shown in Fig EV 5C, J.

Comment 4: Fig 6 The Y axis labels for many figures are too small.

Response 4: The Y axis labels in Fig 6 have been enlarged.

Comment 5: Fig 7 The labels in B and C are very hard to read. The gating strategy for TIL should be shown.

Response 5: We have made the labels in Fig 7B and C clearer. We have shown the gating strategy for TIL CD4 and CD8 T cells in Fig EV 5D; the gating strategy for the TIL Tregs was from the gated CD4⁺ T cells.

Review #2:

General Comment: In this manuscript, Lin and colleagues report the identification of TAPBPL. They propose that TAPBPL is a novel T cell co-inhibitory molecule. To support their claims, the authors present data showing that TAPBPL can be expressed on different immune and tumor cells and, importantly, that manipulation of TAPBPL signaling *in vivo* in EAE and cancer can be performed for therapeutic purposes. Given the importance of co-inhibitory receptors, notably in the cancer field, this study could be relevant in the context of cancer treatment. There are however issues with the study presented that limit the translational potential of the work.

Response to General Comment: We thank the reviewer for his/her thoughtful and positive evaluation of our study and the comments that have improved the manuscript.

Comment 1: In places, the doses of stimulating agents used to study TAPBPL expression are very high. This applies for instance on the doses of IFN γ used (Supplementary Figure 1) and also, even more importantly, on the doses of LPS used. It is curious why the authors need to use such high doses to achieve expression. Showing the expression levels of control molecules (for instance co-stimulation molecules on DC) may help to clarify the relevance of the proposed biological effects.

Response 1: We are sorry that there was a typo for the dose of IFN γ (the 20 $\mu\text{g/ml}$ should be 20 ng/ml) in previous Supplemental Figure 1 (now Fig EV1G, H). We have corrected the typo. We used 2.5-10 $\mu\text{g/ml}$ LPS in our studies (Fig EV1A). Other investigators have also used 2.5-10 $\mu\text{g/ml}$ dose range of LPS in their studies (JEM 2001, 193:839; Nature Immunology 2003, 4:670). We have shown the co-expression of PD-L1 and TAPBPL on the immune cells after stimulation with LPS (Fig EV1C).

Comment 2: It is not clear how statistics were computed in the whole manuscript.

Response 2: We have added the statistical method in the paper.

Comment 3: The effects on T cell proliferation are shown and convincing. However, the mechanism of action of TAPBPL signaling remains elusive. Effects on T cell activation should be explored. What about cytokine release?

Response 3: We have performed the studies to determine the mechanism of action of TAPBPL signaling and shown that TAPBPL and PD-L1 affect different TCR signaling molecules (Fig EV3). The effects of TAPBPL-Ig on cytokine production from T cells *in vitro* have been shown in new Appendix Figure S3.

Comment 4: It remains difficult to understand from *in vivo* experiments how signaling induced by TAPBPL works. What are the cells accounting for the observed effects in EAE? In cancer? The authors discuss that Tregs may be involved but in the context of this study, this needs to be investigated in more depth.

Response 4: We have shown that deletion of Tregs partly abolished the effect of TAPBPL-Ig on EAE (Fig 6R), indicating that Tregs play a role in the observed effect in EAE. In addition, we have demonstrated that TAPBPL-Ig increases the differentiation of naïve T cells into Tregs *in vitro* (new Fig EV4D, E). Furthermore, we have shown that depletion of CD8 T cells partly

abrogated the antitumor effect of the anti-hTAPBPL mAb (Fig EV5I), suggesting that CD8 T cells, at least in part, mediated the antitumor activity of the anti-hTAPBPL mAb.

Reviewer #3:

General Comment: The authors reported a novel molecule TAPBPL which showed inhibitory function on T cell activation. The study was well logically designed, the data are overall convincing. Some technical issues need to be addressed.

Response to General Comment: We thank the reviewer for his/her thoughtful and positive evaluation of our study and the comments that have improved the manuscript.

Comment 1: The expression of TAPBPL on various immune cells seem not related to their activation status, which is different from many B7 family members. This should be validated with other methods, such as RT-PCR. Can type I or II IFN stimulation upregulate TAPBPL on dendritic cells or macrophages?

Response 1: We have shown the expression levels of TAPBPL protein on dendritic cells and monocytes were increased upon activation by LPS or IFN- γ (Fig 2A and B and Fig EV1A). The data have been validated by qRT-PCR, showing that the TAPBPL mRNA expression levels in the immune cells were consistent with the protein expression levels (Fig EV1B).

Comment 2: Staining of TAPBPL on tumor tissues are not clear. What substrate was used for TAPBPL visualization? How is the section counterstained? In addition, additional methods, such as RT-PCR, are suggested to confirm the results.

Response 2: The sections were developed with the ImmPACT DAB Peroxidase Substrate and Counterstained with hematoxylin (from Vector Laboratories). We have used qRT-PCR to confirm the expression levels of the TAPBPL mRNA in cancer cells (Fig EV1F).

Comment 3: Figure 4 and Figure 5 can be inverted, since T cell activation is before T cell proliferation.

Response 3: Thank you for the suggestion. We have reversed the previous Figure 4 and Figure 5.

Comment 4: The increased Treg population in EAE model upon hTAPBPL-Ig treatment and reduced Treg in tumor model upon anti-hTAPBPL treatment are interesting. Is this a direct or indirect effect of hTAPBPL manipulation? Can hTAPBPL-Ig increase Treg differentiation *in vitro*?

Response 4: We have provided new data showing that hTAPBPL-Ig increases the differentiation of naïve T cells into Tregs *in vitro* (Fig EV4D, E), indicating that hTAPBPL can directly affect Tregs.

Comment 5: The authors confirmed the neutralizing role of anti-hTAPBPL on hTAPBPL-Ig's inhibitory effect. This is not surprising. Can anti-hTAPBPL increase T cell activation/proliferation? This would be very informative to be shown at least using *in vitro* assay.

Response 5: Although the anti-hTAPBPL mAb neutralizes proliferation and activation inhibitory activity of hTAPBPL on T cells (Figure 7A-C), as indicated in our response to comment #3 from reviewer #1, this mAb alone did not increase the activation and proliferation of purified T cells *in vitro* (Fig EV5A, B).

Comment 6: The authors use human TAPBPL related reagents for mouse study. Can mTAPBPL-Ig or anti-mTAPBPL demonstrate more clear-cut results? This is highly expected.

Response 6: We have provided new data showing that mTAPBPL-Ig, when used at lower concentrations than hTAPBPL-Ig (0.8-1.6 vs 5-15 $\mu\text{g/ml}$), could inhibit the activation and proliferation of mouse CD4 and CD8 T cells (Appendix Fig S2).

5th Mar 2021

Dear Dr. Lai,

Thank you for the submission of your revised manuscript to EMBO Molecular Medicine. I am pleased to inform you that we will be able to accept your manuscript pending the following final amendments:

- 1) Please address all the concerns raised by the referee #2. Statistical analyses should be repeated using appropriate statistical tests. Regarding the second point, no new experiments are required, instead please tone down the conclusions regarding the CD8 T cell contribution and in vivo role of TAPBPL.
- 2) In the main manuscript file, please do the following:
 - Correct/answer the track changes suggested by our data editors by working from the attached/uploaded document.
 - Remove text highlight colour.
 - Add callouts for EV Fig 3A and 3B.
 - Make sure that all special characters display well.
 - In M&M, provide the antibody dilutions that were used for each antibody.
 - In M&M, the statistical paragraph should reflect all information that you have filled in the Authors Checklist, especially regarding randomization, blinding, replication.
 - Indicate in legends exact $n=$ and exact $p=$ values, not a range, along with the statistical test used. To keep the figures "clear" some authors found providing an Appendix table Sx with all exact p -values preferable. You are welcome to do this if you want to.
 - Correct the reference citation in the reference list. Where there are more than 10 authors on a paper, 10 will be listed, followed by "et al.". Please check "Author Guidelines" for more information. <https://www.embopress.org/page/journal/17574684/authorguide#referencesformat>
- 3) Source data: Please upload one file per figure and name it Source Data file for Fig 1 etc. Please label western blots in source data and all other source data.
- 4) Appendix: Please add table of content.
- 5) For more information: There is space at the end of each article to list relevant web links for further consultation by our readers. Could you identify some relevant ones and provide such information as well? Some examples are patient associations, relevant databases, OMIM/proteins/genes links, author's websites, etc...
- 6) As part of the EMBO Publications transparent editorial process initiative (see our Editorial at <http://embomolmed.embopress.org/content/2/9/329>), EMBO Molecular Medicine will publish online a Review Process File (RPF) to accompany accepted manuscripts. This file will be published in conjunction with your paper and will include the anonymous referee reports, your point-by-point response and all pertinent correspondence relating to the manuscript. Let us know whether you agree with the publication of the RPF and as here, if you want to remove or not any figures from it prior to publication. Please note that the Authors checklist will be published at the end of the RPF.
- 7) Please provide a point-by-point letter INCLUDING my comments as well as the reviewer's reports and your detailed responses (as Word file).

I look forward to reading a new revised version of your manuscript as soon as possible.

Yours sincerely,

Zeljko Durdevic

***** Reviewer's comments *****

Referee #2 (Remarks for Author):

In this revised version of their work authors have attempted to address the concerns from the referees.

I still have concerns regarding the work in its current form:

- 1) In multiple places, authors rely on T-tests to compare more than two groups. This is not appropriate. In addition, statistics should also take into account the fact that multiple comparisons have been made. In the present case, some differences shown are minor, and it is important to ensure these differences have biological and statistical significance.
- 2) Authors suggest that Tregs and CD8 T cells contribute to their observations, but their explanations are very brief. The control group CD8 T cell depletion alone is missing in the in vivo experiment. The contribution of Tregs and CD8 T cells are in the end relatively modest according to the data shown. The role of TAPBPL in vivo remains unclear.

Referee #3 (Remarks for Author):

I have no more concerns.

The authors performed the requested editorial changes.

Comments from Referee #2:

Comment 1: In multiple places, authors rely on T-tests to compare more than two groups. This is not appropriate. In addition, statistics should also take into account the fact that multiple comparisons have been made. In the present case, some differences shown are minor, and it is important to ensure these differences have biological and statistical significance.

Response 1: As indicated in our response to editor's comment 1, we have repeated statistical analyses using appropriate statistical tests. For comparing means of 2 groups, we have used Student's *t*-test. For comparing means of multiple groups, we have used one-way or two-way ANOVA followed by Dunnett test or Tukey test.

Comment 2: Authors suggest that Tregs and CD8 T cells contribute to their observations, but their explanations are very brief. The control group CD8 T cell depletion alone is missing in the in vivo experiment. The contribution of Tregs and CD8 T cells are in the end relatively modest according to the data shown. The role of TAPBPL in vivo remains unclear.

Response 2: In the CD8 T cell depletion (Figure EV5I), we have used two controls: hTAPBPL isotype Ab and anti-CD8 isotype Ab. As indicated in our response to editor's comment 1, we have also toned down the conclusions regarding the CD8 T cell contribution and in vivo role of TAPBPL.

We would like to thank the editors and reviewers for their insightful and helpful comments of our revised paper. We have attempted to respond to all their comments, as indicated below and in the revised manuscript. We hope that these revisions are satisfactory.

Comments from Editors:

Comment 1: Please address all the concerns raised by the referee #2. Statistical analyses should be repeated using appropriate statistical tests. Regarding the second point, no new experiments are required, instead please tone down the conclusions regarding the CD8 T cell contribution and in vivo role of TAPBPL.

Response 1: We have repeated statistical analyses using appropriate statistical tests. For comparing means of 2 groups, we have used Student's *t*-test. For comparing means of multiple groups, we have used one-way or two-way ANOVA followed by Dunnett test or Tukey test. We have also toned down the conclusions regarding the CD8 T cell contribution and in vivo role of TAPBPL.

Comment 2: In the main manuscript file, please do the following:

- Correct/answer the track changes suggested by our data editors by working from the attached/uploaded document.
- Remove text highlight colour.
- Add callouts for EV Fig 3A and 3B.
- Make sure that all special characters display well.
- In M&M, provide the antibody dilutions that were used for each antibody.
- In M&M, the statistical paragraph should reflect all information that you have filled in the Authors Checklist, especially regarding randomization, blinding, replication.
- Indicate in legends exact $n=$ and exact $p=$ values, not a range, along with the statistical test used. To keep the figures "clear" some authors found providing an Appendix table Sx with all exact p -values preferable. You are welcome to do this if you want to.
- Correct the reference citation in the reference list. Where there are more than 10 authors on a paper, 10 will be listed, followed by "et al.". Please check "Author Guidelines" for more information.

Response 2: We have removed text highlight color and added callouts for EV Fig 3A and 3B. We have made sure that all special characters display well. In M&M, we have provided the antibody dilutions that were used for each antibody. The statistical paragraph in M&M has reflected all information that we have filled in the Authors Checklist. We have indicated in legends exact $n=$ and the statistical test used; the exact p -values are listed in Appendix Table 1. We have also corrected the reference citation in the reference list.

Comment 3: Source data: Please upload one file per figure and name it Source Data file for Fig 1 etc. Please label western blots in source data and all other source data.

Response 3: We have uploaded one file of source data per figure and name it according the instruction. We have also labeled western blots in the source data.

We are pleased to inform you that your manuscript is accepted for publication and is now being sent to our publisher to be included in the next available issue of EMBO Molecular Medicine.

Corresponding Author Name: Lajun Lai
Journal Submitted to: EMBO Molecular Medicine
Manuscript Number: EMM-2020-13404